# Simulation of Soil Organic Carbon Content Based on Laboratory Spectrum in the Three-Rivers Source Region of China

**Wei Zhou** [1,2], **Haoran Li** [1,3,*], **Shiya Wen** [1], **Lijuan Xie** [3], **Ting Wang** [1], **Yongzhong Tian** [1] **and Wenping Yu** [1]

[1] Chongqing Jinfo Mountain Karst Ecosystem National Observation and Research Station, School of Geographical Sciences, Southwest University, Chongqing 400715, China; zw20201109@swu.edu.cn (W.Z.); meinwsy@email.swu.edu.cn (S.W.); swuwt230@email.swu.edu.cn (T.W.); tyzlf@swu.edu.cn (Y.T.); ywpgis2005@swu.edu.cn (W.Y.)

[2] State Key Laboratory of Resources and Environmental Information System, Institute of Geographic Sciences and Natural Resources Research, CAS, Beijing 100101, China

[3] Department of Geography and Land and Resources, Chongqing Jiaotong University, Chongqing 400074, China; xielijuan@piesat.cn

[*] Correspondence: lhr97@mails.cqjtu.edu.cn

**Abstract:** Soil organic carbon (SOC) changes affect the land carbon cycle and are also closely related to climate change. Visible-near infrared spectroscopy (Vis-NIRS) has proven to be an effective tool in predicting soil properties. Spectral transformations are necessary to reduce noise and ensemble learning methods can improve the estimation accuracy of SOC. Yet, it is still unclear which is the optimal ensemble learning method exploiting the results of spectral transformations to accurately simulate SOC content changes in the Three-Rivers Source Region of China. In this study, 272 soil samples were collected and used to build the Vis-NIRS simulation models for SOC content. The ensemble learning was conducted by the building of stack models. Sixteen combinations were produced by eight spectral transformations (S-G, LR, MSC, CR, FD, LRFD, MSCFD and CRFD) and two machine learning models of RF and XGBoost. Then, the prediction results of these 16 combinations were used to build the first-step stack models (Stack1, Stack2, Stack3). The next-step stack models (Stack4, Stack5, Stack6) were then made after the input variables were optimized based on the threshold of the feature importance of the first-step stack models (importance > 0.05). The results in this study showed that the stack models method obtained higher accuracy than the single model and transformations method. Among the six stack models, Stack 6 (5 selected combinations + XGBoost) showed the best simulation performance (RMSE = 7.3511, $R^2$ = 0.8963, and RPD = 3.0139, RPIQ = 3.339), and obtained higher accuracy than Stack3 (16 combinations + XGBoost). Overall, our results suggested that the ensemble learning of spectral transformations and simulation models can improve the estimation accuracy of the SOC content. This study can provide useful suggestions for the high-precision estimation of SOC in the alpine ecosystem.

**Keywords:** soil organic carbon; visible-near infrared spectroscopy; characteristic band; extreme gradient boosting; Tibetan plateau

## 1. Introduction

Soil organic carbon (SOC) is well known to have a significant impact on global carbon cycling, soil quality and environmental protection [1]. Traditionally, methods for measuring SOC content are time-consuming, requiring intensive workloads and high research costs. At present, with a greater understanding of the relationship between the spectral reflectance of soil and the spectral response of organic matter, hyperspectral retrieval can provide a basis for the rapid and effective monitoring of SOC content [2]. Yet, the most effective combination of retrieval models and spectral transformations has not been identified for Alpine ecosystems. Here, the retrieval models of SOC content may be divided into the

categories of linear and nonlinear models. The linear regression models include multiple linear regression (MLR), principal component regression, and partial least squares regression (PLSR) [2–5]. Moreover, machine learning methods are widely used in SOC content spectral retrieval and include methods such as support vector machine (SVM), artificial neural network, random forest (RF) and boosted regression tree, etc. [6–9].

SVM has a good effect in field spectral simulation of SOC content in farmland [10]. Some scholars have used CARS (competitive adaptive reweighted sampling) in combination with RF models and achieved a better prediction effect than that obtained using PLSR [11]. DNN (deep learning neural networks) resulted in the best modeling accuracies, followed by RF, XGBoost (extreme gradient boosting), ANN [12], and Cubist [13]. In previous studies, we found that the retrieval accuracy of RF and SVM is higher than that of PLSR [14]. Here, PLSR, RF and XGBoost are used to build simulation models and compare their accuracies in SOC content estimation.

As the noise and error produced in the spectral measurement process are particularly complex, the spectral pre-processing method is an important prerequisite for improving the retrieval accuracy of SOC content. Preprocessing is used to reduce the noise of the spectrum acquisition and extract more characteristic band information. Currently, smoothing, multiple scattering correction (MSC), baseline correction, standard orthogonal transform, difference transform, logarithmic reciprocal (LR) transform, wavelet transform, continuous media removal, and wavelet packet analysis are widely used [15–17]. Previous studies have found that MSC, baseline correction, and differential transformation can effectively improve retrieval accuracy of SOC content [18]. Additionally, differential transformation is very sensitive to the spectral signal-to-noise ratio, and so the first-order differential (FD) transform is more prominent in this respect. In contrast, the second-order differential (SD) transform is more effective in eliminating the baseline drift and background effects of some instruments [19]. The LR transform can reduce the influence of random factors caused by light conditions and changes in terrain and enhance the spectral differences of visible-light areas [17]. Continuum removal (CR) can effectively highlight the absorption and reflection characteristics of the spectrum and, in doing so, improve the correlation between the spectral data and organic carbon. MSC is a preprocessing method used to separate the spectral scattering signal from the chemical absorption information, which can eliminate the spectral differences of the same batch of samples caused by the uneven soil sample particles during the diffuse reflection process [20].

Above all, the different spectral transformation methods and estimation models showed different advantages in previous studies. Recently, stacking in ensemble learning based on models has been proved to improve the prediction accuracy of SOC content using environment variables [13]. Some studies use models averaging combined soil properties or weighted models averaging to improve the mapping soil properties [21,22]. Because the ensemble method is considered the most promising framework to ensure the robustness of variable subset in conditions of high dimensional dataset and low-size sample [23,24]. The above studies merely combined multiple models to reduce the limitations of a single model. However, the various spectral pre-processing transformation methods can be combined and display the potential of improving the performance of SOC content estimation. Therefore, in this study, we will explore the ensemble learning method of spectral transformation methods and estimation models.

The Three-Rivers Source Region (TRSR), located in the hinterland of the Qinghai Tibet Plateau, is an ecologically fragile region particularly sensitive to impacts of global climate change. As global temperatures rise and human disturbances intensify, the degradation of grassland and the severe desertification of land in this region has brought about temporal and spatial changes to the SOC content [25]. The TRSR is characterized by areas of permafrost and seasonal permafrost. Therefore, a warming climate may induce environmental changes that accelerate the microbial breakdown of organic carbon, thereby releasing carbon dioxide. In this way, a positive feedback cycle is produced that can further accelerate

climate change [26]. On the other hand, warming can also increase primary production, thereby increasing soil carbon stocks [27].

Meanwhile, overgrazing might accelerate carbon decomposition rates and shorten the turnover time by shifting soil microbial composition [28]. With the aggravation of grassland degradation, SOC content has significantly decreased in the Tibetan alpine meadow [29]. Therefore, SOC content estimation in this alpine ecosystem is vital to the sequestration of soil carbon, along with its feedback to climate change.

To estimate SOC content and explore the optimal ensemble learning method of spectral transformations and simulation models, we conducted a four-year field sampling study and obtained 272 soil samples. We then measured SOC content and soil spectral data for these samples in the laboratory. Moreover, the laboratory spectrum of dry soil can be used to correct the spectrum of the field in situ soil spectrum or those of hyperspectral imaging in future studies [30]. Additionally, as different combinations of spectral transformations and models have been shown to achieve different simulation accuracy [14], the explorations of the optimal ensemble learning method are crucial for future SOC content monitoring using remote sensing images.

The objectives of this study were (1) to reveal the topsoil SOC content characteristics of alpine grassland ecosystems using the field soil samples in the TRSR; (2) to explore and define the best ensemble learning methods of spectral transformations and estimation models for SOC content. The results of this study may therefore provide a reference for the estimation and monitoring of SOC content in alpine areas.

## 2. Materials and Methods

### 2.1. Study Area

The Three-Rivers Source Region (TRSR) is located in the southern part of Qinghai Province, at N31°39′–36°12′, E89°45′–102°23′. The administrative area includes 16 counties in Yushu, Guoluo, and Tibetan Autonomous Prefectures of Huangnan and Tanggula Townships in Golmud City, with a total area of 302,000 km$^2$. TRSR is part of the Tibetan Plateau and consists of polar tundra climate (ET) and dry belt steppe climate (Bs) in the Köppen Geiger classification system. The area exhibits a small annual temperature difference and a large daily temperature difference. The duration of sunshine in the TRSR is long, with intense radiation and no distinct seasonal variation. Overall, the study region is semi-arid, with <400 mm of precipitation annually.

The TRSR is dominated by mountains with complex terrain, and its altitude ranges between 3335–6564 m. Overall, the topography of the central and northern portions of the region exhibits little fluctuation, with many wide and flat areas. Due to this flat terrain, in combination with its long history of glaciation and poor drainage, a large area of wetland has been formed. The southeast area of the study region mainly includes high mountains and valleys, with strong geomorphic cutting and steep terrain. The vertical zonal distribution law of soil and vegetation is determined by these mountainous landforms and climate conditions. The vegetation types in the area include temperate grasslands, temperate desert grasslands, alpine meadows, alpine grasslands, and alpine meadow grasslands (Figure 1). The soil types are alpine cold desert soil, alpine meadow soil, alpine grassland soil, mountain meadow soil, gray cinnamon soil, chestnut soil and mountain forest soil, in which alpine meadow soil is the main soil. Additionally, swampy meadow soil is also widely distributed. However, the soil in the TRSR has a short development time, creating a thin soil layer with coarse soil texture and poor water holding capacity.

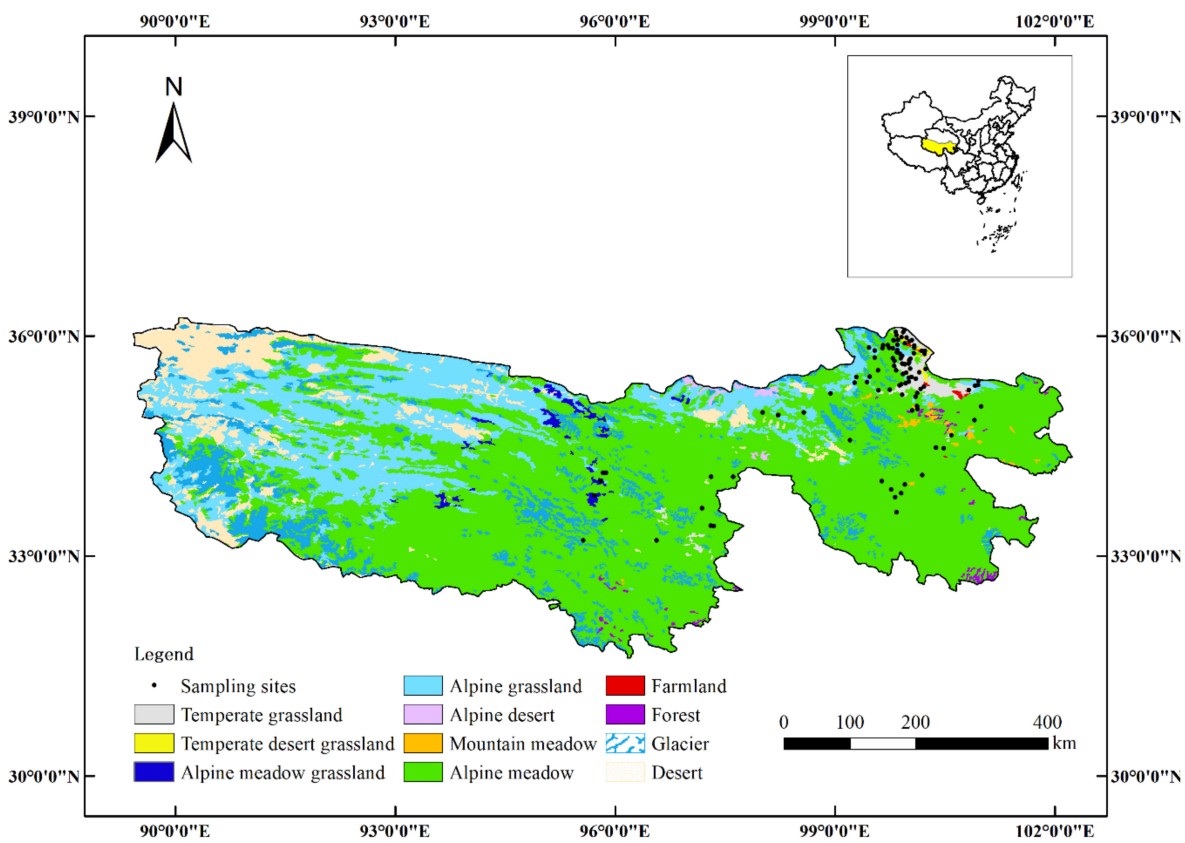

**Figure 1.** Distribution of soil sampling sites and different vegetation types in the Three-Rivers Source Region.

The TRSR is China's largest nature reserve and is one of the most concentrated areas of high-altitude biodiversity globally, mainly due to its unique landforms and vegetation types. However, under the influence of global climate change and human activities, the area has experienced serious degradation and has become fragile [31]. This process has ultimately influenced the terrestrial carbon cycle and soil carbon sequestration ability of the area. Therefore, the ability to accurately estimate SOC content is crucial for the alpine ecosystem carbon sequestration and the implementation of effective ecosystem management strategies. In this study, the sampling sites are mainly located in the eastern and central portions of the TRSR, considering factors such as accessibility and poor natural conditions. The number of sampling sites and soil types of the different land-use types are shown in Table 1.

**Table 1.** Numbers of samples and soil type for each vegetation type.

| Vegetation Type | Sample Number | Elevation (m) | Soil Type |
|---|---|---|---|
| Alpine meadow | 101 | 3200–4000 | Grass felt soil, Dark felt soil, |
| Alpine grassland | 36 | 3400–4200 | Grass felt soil, Chestnut soil |
| Alpine meadow grassland | 46 | 3700–4400 | Meadow soil |
| Swamp meadow | 8 | 4500–4700 | Swamp soil |
| Temperate grassland | 59 | 3100–3200 | Chestnut soil |
| Desert grassland | 6 | 2900–3000 | Chestnut soil |
| Farmland | 13 | 2600–3000 | Chestnut soil |
| Woodland | 3 | 3390–3692 | Mountain forest soil |

## 2.2. Sample Collection and Preparation

In this study, field surveys were conducted from late July to early August from 2017 to 2020 (Figure 1). Based on the geographical features and limited accessibility of the study area, Google satellite imagery was used to reasonably select sampling points. At each sampling point, a five-point mixed sampling method was used to collect soil samples, with 1 kg of soil being collected for each. The depth of each soil sampling point was 20 cm. After collection, the samples were kept sealed in compact bags and were labeled. All sampling points were precisely positioned using a hand-held GPS. The soil properties, land use types, and vegetation cover conditions for each sampling point were recorded in detail. The sampling areas were photographed by a digital camera, and the spatial distribution map of the sampling points was drawn. In total, 272 top-soil samples (0–20 cm) were collected over different land use types and were taken as the research object.

The collected soil samples were placed in a cool and ventilated room to air-dry. Any sand, gravel, roots, or leaf residues were removed from the soil sample before being ground in a ceramic grinding bowl. The samples were then sieved (0.25 mm). Finally, the samples were divided into three parts, two for indoor chemical analysis and soil spectrum determination, and one was sealed and stored for future reference to prevent cross-contamination.

## 2.3. Determination of Soil Spectrum and SOC Content

A certain amount of oxidant (potassium dichromate sulfuric acid solution) was used to oxidize the organic matter in the soil under heat. The remaining oxidant was titrated with ammonium ferrous sulfate. Finally, the amount of oxidant consumed was used to calculate the soil organic carbon content (SOC, g/kg). The formula is as follows:

$$\text{SOC} = \frac{c \times (V_0 - V) \times 0.003 \times 1.10}{m} \times 1000 \tag{1}$$

where $V_0$ is the volume of ferrous sulfate standard solution consumed in blank test (mL); V is the volume of ferrous sulfate standard solution consumed in test (mL); c is the concentration of ferrous sulfate standard solution (mol/L); 0.003 is the millimolar mass of a quarter carbon atom (g); 1.10 is the coefficient of oxidation correction; m is the mass of the dried sample (g); and 1000 is a conversion to content per kilogram. The parallel measurement results are expressed by arithmetic mean, and the significant three-dimensional figures are retained.

Chemical analysis results of SOC content and other indicators of selected 10 samplings are shown in Table 2.

**Table 2.** Chemical analysis results of SOC content and other indicators.

| Number | m (g) | c (mol/L) | $V_0$ (mL) | V (mL) | SOC (g/kg) |
|--------|-------|-----------|------------|--------|------------|
| 1 | 0.2037 | 0.20 | 20.60 | 12.1400 | 27.4087 |
| 2 | 0.1503 | 0.20 | 20.60 | 8.8900 | 51.4170 |
| 3 | 0.1991 | 0.20 | 20.60 | 12.8100 | 25.8211 |
| 4 | 0.2038 | 0.20 | 20.60 | 10.4000 | 33.0297 |
| 5 | 0.2006 | 0.20 | 20.60 | 13.1800 | 24.4108 |
| 6 | 0.1955 | 0.20 | 20.60 | 11.9000 | 29.3685 |
| 7 | 0.2036 | 0.20 | 20.60 | 11.3200 | 30.0801 |
| 8 | 0.2009 | 0.20 | 20.60 | 13.7000 | 22.6662 |
| 9 | 0.2018 | 0.20 | 20.60 | 16.1100 | 14.6837 |
| 10 | 0.2050 | 0.20 | 20.60 | 13.6100 | 22.5026 |

The soil spectrum reflectance was measured based on the dry soil samples in the absence of light using the Analytical Spectral Devices (ASD) FieldSpec 4 portable spectrometer. The spectral detection range was 350–2500 nm. The interior spectral measurement conditions were as follows: A 50 W halogen lamp was selected as the light source. The zenith angle of the light source was 45°, the distance from the light source to the soil surface

was 30 cm, and the field of view of the bare optical fiber probe was 25°. Additionally, the instrument was preheated for 30 min before use, the whiteboard was calibrated, and the experiment was conducted after the instrument was stable. The soil sample was placed in a dish with 100 mm diameter and 1.5 mm height. The black swan flannelette was taken as the background, and the optical fiber probe was positioned 15 cm above the soil sample. Each soil sample was measured in four directions, and five soil spectral curves were recorded for each direction. The sampling interval was 1 nm with 20 samples collected. Finally, the average value was selected as the spectral reflectance of the sample.

### 2.4. Pre-Processing Transformations

To reduce the influence of noise on the soil spectrum, pre-processing was conducted in turn, namely band clipping, Savitzky–Golay smooth filtering (S-G smooth), multiple scattering correction (MSC), logarithmic reciprocal (LR), continuum removal (CR), and first-order derivative (FD).

Initially, the band clipping was mainly focused on the bands where the spectrometer generates large amounts of noise (350–399 nm and 2451–2500 nm). After clipping, 2051 bands with a wavelength range of 400–2450 nm were retained.

Second, to avoid the influence of edge noise, the Savitzky–Golay filtering method was used to smooth the spectral curve. S-G filtering is widely used in data smoothing and denoising and can effectively remove noise and preserve the overall characteristics of the spectral curve [32], which in turn can improve the estimation ability [33]. To highlight the spectral band and separate the parallel background values, eight kinds of spectral transformations were performed on the smoothed soil original spectral curve, namely the S–G smooth, MSC, LR, CR, FD, MSC–FD, LR–FD, and CR–FD. Specifically, MSC was applied to remove the multiplicative interferences of scatter and particle size. The basic principle of MSC is to calculate the average spectrum of the sample to establish a linear regression equation and correct the spectral information. CR and LR are also used for continuum removal and are both common spectral analysis methods. Here, the soil spectrum can reduce the influence of background information through de-enveloped processing, so as to more effectively highlight the absorption and reflection characteristics of the spectral curve.

The first derivative (FD) and second derivative (SD) of reflectance are utilized to remove baseline drift and background interferences, as well as to resolve overlapped spectra, although the noise level increases significantly in higher-order derivative calculations [34]. The application of the first and second derivatives of reflectance in soil properties estimation depends on the quality of the raw spectral data. Previous studies employed the FD [35,36], while others preferred the SD method [37]. Furthermore, some studies found that the FD obtained a better performance effect than SD [38]. The S–G smooth, FD, MSC, and LR were implemented using Unscrambler software, and CR was conducted using ENVI software. The eight spectral transformations are displayed in Figure 2.

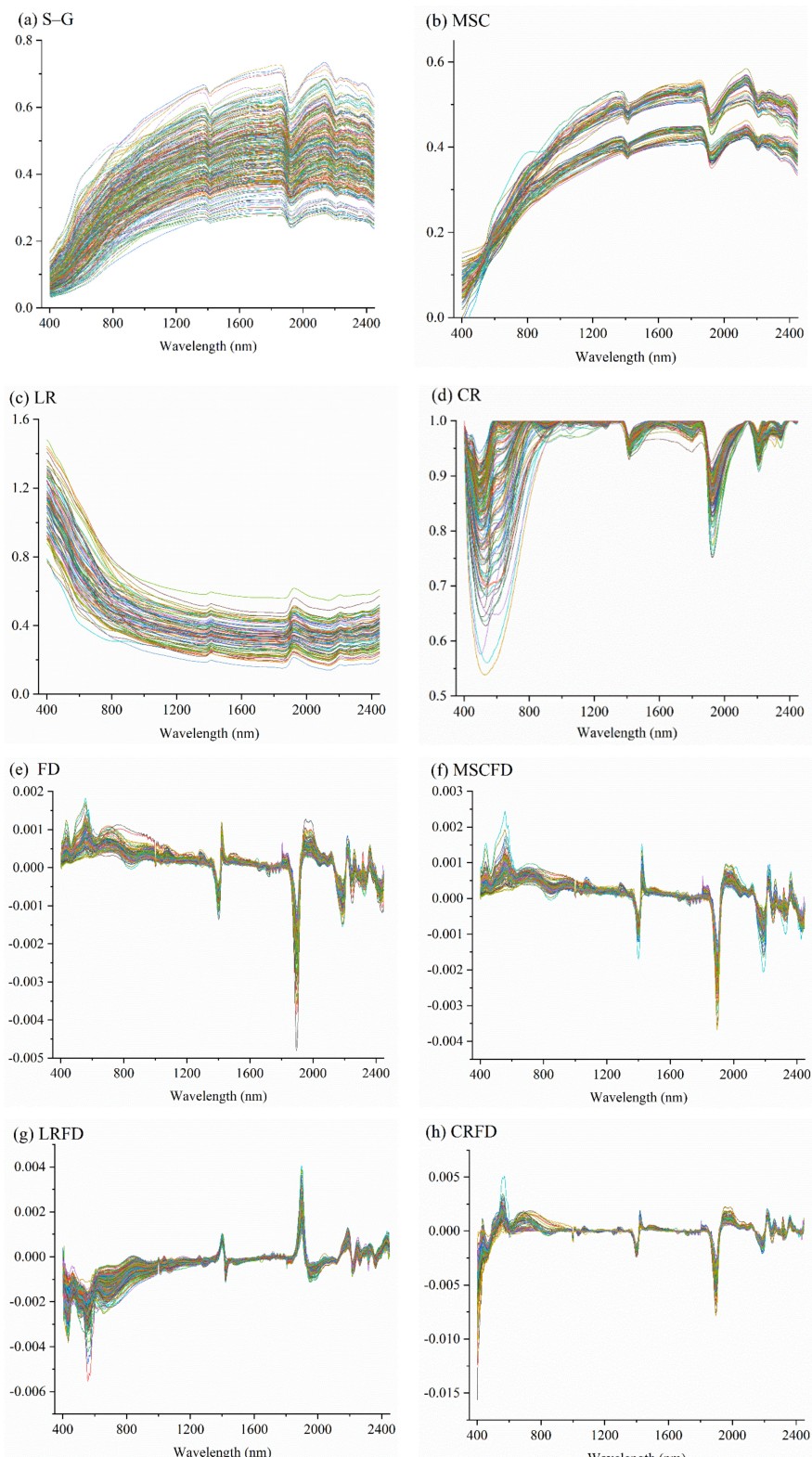

**Figure 2.** Soil spectral curve of eight kinds of spectral transformations. (**a**) Savitzky–Golay smooth filtering, (**b**) multivariate scattering correction, (**c**) logarithmic reciprocal, (**d**) continuum removal, (**d**,**e**) first–order differential, and (**f**–**h**) first–order differential transformation based on (**b**–**d**), respectively. The color lines represented the spectral curve of soil samples.

### 2.5. Hyperspectral Retrieval Models of SOC Content

The original spectrum and reconstructed spectrum were taken as the independent variable, while the SOC content was considered the dependent variable of the models. According to a ratio of 2:1, the modeling samples and verification samples were selected for the model building of SOC content.

#### 2.5.1. Individual Models

The PLSR method has demonstrated a strong ability to solve multicollinearity problems, and as such, is widely used in the field of SOC content hyperspectral retrieval [5,39]. The PLS method combines MLR and principal component analysis (PCA). Compared with the traditional MLR method, PLS can be used for regression modeling in cases where there are a large number of bands and considerable autocorrelation.

SVR (support vector machine regression models) is a statistical learning method on the basis of structural risk minimization and can deal with the problems of small samples, nonlinearity, and high dimension and overcome the difficulties of local minimum in neural networks.

RF is a form of machine learning algorithm used to model the relationships between target variables and potential predictors [40]. RF is widely used in nonlinear and big data applications where it can reduce the computation required while ensuring models accuracy [41]. The RF model considers decision trees as the basic unit and averages the results from all trees to obtain its predicted result. Many decision trees are constructed in RF to ensure the stability of the models, where each tree is independently constructed by a unique bootstrap sample of the training dataset [42]. Further, only a random subset of the covariates is evaluated at each node. For these reasons, RF with a sufficiently large number of trees is robust against overfitting, noise, as well as non-informative and correlated features. This modeling technique is generally preferred in soil properties mapping or SOC content estimation studies because it can estimate the relative importance of variables, is insensitive to overfitting and produces stable and accurate predictions [9].

XGBoost is also a tree-based ensemble method [43], combining the advantages of two algorithms (i.e., regression trees and boosting) to improve the performance of a single model. Boosting is a numerical optimization algorithm that minimizes the loss function by adding a new tree to the previous regression tree model at each step [44]. In this way, the algorithm focuses more on samples with higher uncertainty. Finally, all generated models are combined to calculate the outcome. Therefore, the XGBoost model is useful for SOC content retrieval or soil properties mapping studies.

#### 2.5.2. Stacking Model

At present, meta-learning method is a highly effective method for stacking models. There are two levels in stacking framework of meta-learning method: level 0 is the base model, and level 1 is the prediction model [13]. In principle, the method takes the measured value of the original data set as the dependent variable data set Y, and the output of the base models as the input features, that is, the independent variable data set X, and inputs them together into the regression prediction model to achieve the purpose of integrating features of multiple base models [45]. In addition, the meta-learning method can also be used to filter multiple base models [46].

In this paper, three machine learning models, SVR, RF and XGboost, were selected as prediction models, respectively. And 16 preprocessor-model combinations established by eight spectral transformation methods (SG, MSC, LR, CR, FD, MSCFD, LRFD, CRFD) and two independent models (RF, XGBoost) were stacked to obtain six stack models. The process of stacking models can be seen in Figure 3.

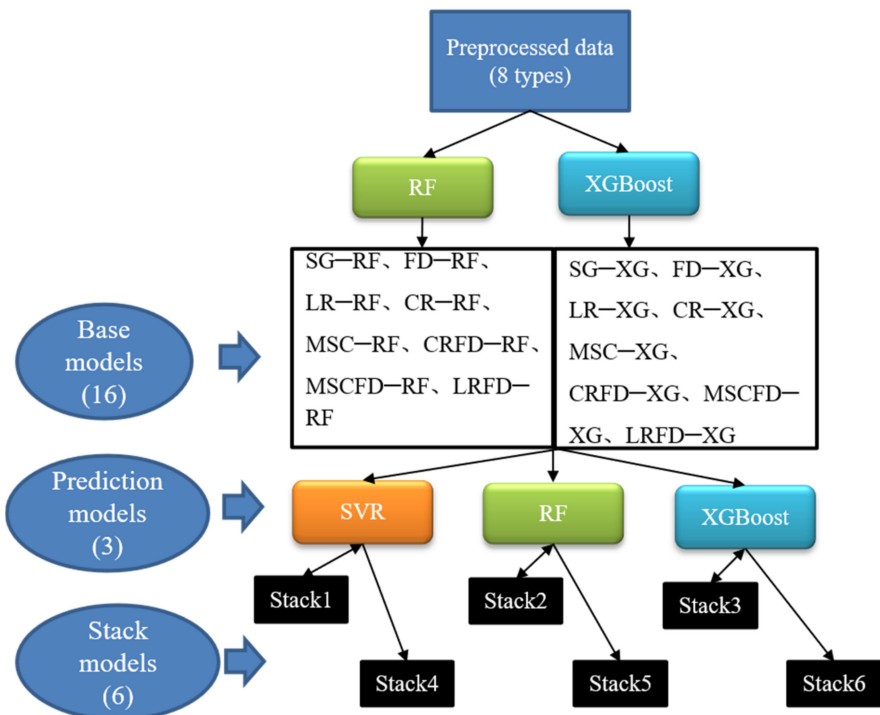

**Figure 3.** Stacking model processes based on three models and eight pre-processing transformations (SVR is support vector machine regression model, RF is the random forest regression model, XGBoost is the extreme gradient tree regression model. Stack1~ stack3 are stacked models based on 16 preprocessor-model combinations. After modeling Stack1~Stack3, stack4~stack6 are stacked models based on the selection of important variables. The SG is Savitzky–Golay smooth filtering, LR is the logarithmic reciprocal, CR is the continuum removal, MSC is the multivariate scattering correction, FD is the first-order differential, and CRFD, MSCFD and LRFD are the first-order differentials based on CR, MSC and LR, respectively).

### 2.6. Modeling Evaluation

In this study, four common performance metrics were used [47], including root mean square error (RMSE), coefficient of determination ($R^2$), the ratio of performance to InterQuartile distance (RPIQ) and the residual prediction deviation (RPD). The $R^2$ value varies between 0 and 1 and indicates the closeness of the observed values to the fitted regression line or the proportion of variance explained by the independent predictors. RMSE indicates the accuracy of the model's prediction. RPIQ compares the interquartile range to the RMSE [48]. An RPD value < 1.0 indicates a very poor model, and in this case, their application is not recommended. Additionally, 1.0 < RPD < 1.4 indicates poor models; 1.4 < RPD < 1.8 indicates fair models; 1.8 < RPD < 2.0 indicates good models, where quantitative predictions are possible; 2.0 < RPD < 2.5 indicates very good quantitative models; and RPD > 2.5 indicates excellent models [49]. Overall, greater $R^2$ and RPIQ values with a lower RMSE indicate better predictive capacity and stability of the models. The calculations for each performance metric are shown below:

$$\text{RMSEP} = \sqrt{\frac{1}{n}\sum_{i=1}^{n}(P_i - O_i)^2} \tag{2}$$

$$R^2 = \left(\frac{\sum_{i=1}^{n}(O_i - \overline{O})(P_i - \overline{P})}{\sqrt{\sum_{i=1}^{n}(O_i - \overline{O})^2}\sqrt{\sum_{i=1}^{n}(P_i - \overline{P})^2}}\right)^2 \tag{3}$$

$$\text{RPIQ} = \frac{Q_3 - Q_1}{\text{RMSEP}} \tag{4}$$

$$RPD = \frac{SD_O}{RMSEP} \tag{5}$$

where Pi and Oi are the predicted and observed SOC content (g/kg) values at the ith location, n is the number of data points, $\overline{P}$ and $\overline{O}$ denote the means for the predicted and observed SOC content, and $Q_1$ and $Q_3$ are the first and third quartiles of observed SOC content, respectively, and $SD_O$ is the standard deviation of observed SOC content.

### 3. Results and Analysis

#### 3.1. Summary Statistics of SOC Content

The average SOC content of the modeling samples and verification samples were all greater than 34 g/kg. Additionally, the coefficient of variation of the samples was relatively close, with all values of moderate variation, indicating that the data distribution was relatively uniform, which therefore met the basic requirements of modeling (Table 3).

**Table 3.** SOC content statistic of modeling sets and verification sets.

| Sample Type | Sample Number | Average (g/kg) | Standard Deviation (g/kg) | Kurtosis | Skewness | Coefficient of Variation (%) |
|---|---|---|---|---|---|---|
| Overall samples | 272 | 34.68 | 22.92 | 3.3 | 1.66 | 66.1 |
| Modeling samples | 182 | 34.9 | 23.31 | 3.39 | 1.67 | 66.81 |
| Verification samples | 90 | 34.23 | 22.22 | 3.3 | 1.64 | 64.92 |

#### 3.2. Correlation Analysis of Soil Spectrum Reflectance and SOC Content

According to the correlation coefficients between SOC content and eight spectral data (original spectrum, FD, MSC, LR, CR, CRFD, MSCFD and LRFD) (Figure 4), both positive and negative correlation band positions between spectrum and SOC content under different transformation forms were observed. The reflectance of the original spectrum (R) was negatively correlated with SOC content. The correlation between them was first observed to increase, then decrease in the wavelength range of 400–2450 nm, and finally reached the peak value near 600 nm. On the contrary, the spectral curve after LR transformation was positively correlated with SOC content. The correlation coefficient peak value appeared at 576 nm, and the overall correlation was higher than the original data. The correlation coefficient between the spectrum and SOC content after the FD treatments fluctuated greatly between both positive and negative values. Compared with the original spectral correlation coefficient, the correlation coefficient of the MSC and CR transformations data in the near-infrared band was enhanced, indicating that the MSC and CR transformations can amplify the absorption characteristics of the original spectrum in the near-infrared band. In addition, the combined transformations of CRFD and MSCFD significantly improved the correlation between SOC content and spectrum reflectance. Therefore, after the spectral transformations, the absorption valleys became amplified to varying degrees, which is of great significance to improving the accuracy of SOC content estimation.

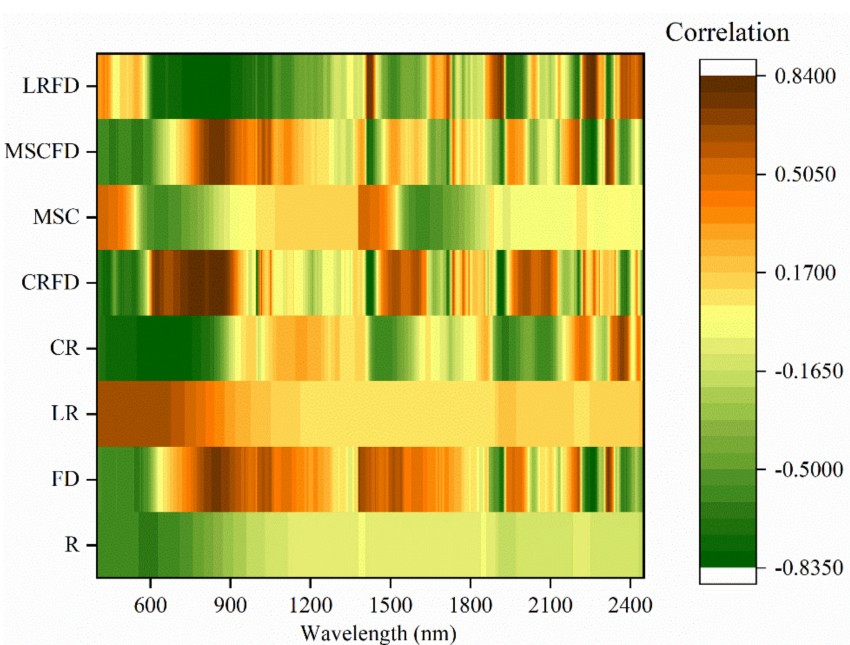

**Figure 4.** The correlation coefficient between SOC content and eight spectral transformations data (SOC is the soil organic carbon, R is the original spectrum merely using Savitzky–Golay smooth filtering, FD is the first-order differential, LR is the logarithmic reciprocal, CR is the continuum removal, MSC is the multivariate scattering correction, and CRFD, MSCFD, and LRFD are the FD based on CR, MSC and LR, respectively).

Moreover, the correlation coefficient reached 0.84 at 800–900 nm (Figure 3), which is related to electronic transitions in iron oxides [50,51]. Peaks near 1400 nm are associated with water hydroxyl (–OH) groups adsorbed to soil organic matter. Band peak of 2200–2300 nm was observed, and the correlation coefficient reached 0.83, which correspond to spectral absorption features influenced by clay, minerals, and organic matter [52,53].

*3.3. Comparison of SOC Content Modeling Results Based on Single Model*

Following a 2:1 ratio, the total sample was divided into the modeling and verification sets. The estimation models (PLSR, RF, and XGBoost) of SOC content were built using the SOC content and the full bands after applying eight spectral transformations (R, LR, CR, MSC, FD, LRFD, CRFD, and MSCFD).

The simulation accuracy results shown in Table 4 are based on the single spectral transformations method. Of the three models compared, the XGBoost model resulted in the best modeling accuracies, followed by RF and PLSR (Table 4). Out of the eight spectral transformations, CR, FD, CRFD, LRFD and MSCFD exhibited better modeling accuracies than the other transformation methods. Of the 24 transformation-model combinations compared, the verification set models accuracy of MSCFD-XGBoost combination was identified as the best ($R^2$ = 0.8966, RMSE = 7.7869, RPD = 2.8452, RPIQ = 3.1521), followed by FD-XGBoost ($R^2$ = 0.8902, RMSE = 7.805, RPD = 2.8386, RPIQ = 3.1448), LRFD- XGBoost, LRFD-RF CRFD-RF.

**Table 4.** SOC content simulation accuracy comparison based on different spectral transformations and models.

| Transformations | Models | Modeling Sets | | Verification Sets | | | |
|---|---|---|---|---|---|---|---|
| | | RMSE | $R^2$ | RMSE | $R^2$ | RPD | RPIQ |
| R | PLSR | 11.1135 | 0.771 | 9.9241 | 0.8089 | 2.2325 | 2.4733 |
| | RF | 5.796 | 0.9496 | 13.3588 | 0.6413 | 1.6585 | 1.8374 |
| | XGBoost | 1.5866 | 0.9956 | 16.1217 | 0.5456 | 1.3743 | 1.5225 |
| LR | PLSR | 11.1135 | 0.771 | 9.9241 | 0.8089 | 2.2325 | 2.4733 |
| | RF | 5.796 | 0.9496 | 13.3588 | 0.6413 | 1.6585 | 1.8374 |
| | XGBoost | 1.5866 | 0.9956 | 16.1217 | 0.5456 | 1.3743 | 1.5225 |
| CR | PLSR | 8.8811 | 0.8538 | 9.2152 | 0.8468 | 2.4042 | 2.6636 |
| | RF | 3.8146 | 0.9789 | 8.0446 | 0.8728 | 2.7541 | 3.0511 |
| | XGBoost | 0.4813 | 0.9996 | 8.2032 | 0.872 | 2.7008 | 2.9921 |
| MSC | PLSR | 12.1779 | 0.7251 | 10.8348 | 0.7693 | 2.0448 | 2.2654 |
| | RF | 4.5176 | 0.9714 | 9.9916 | 0.7994 | 2.2174 | 2.4566 |
| | XGBoost | 0.7942 | 0.9989 | 10.1022 | 0.8055 | 2.1931 | 2.4297 |
| FD | PLSR | 10.8081 | 0.7835 | 9.5417 | 0.8208 | 2.322 | 2.5724 |
| | RF | 3.5427 | 0.9824 | 7.8371 | 0.8755 | 2.827 | 3.1319 |
| | XGBoost | 1.8435 | 0.9939 | 7.805 | 0.8902 | 2.8386 | 3.1448 |
| CRFD | PLSR | 7.1537 | 0.9051 | 8.4846 | 0.8699 | 2.6113 | 2.8929 |
| | RF | 4.0651 | 0.974 | 7.9954 | 0.8797 | 2.771 | 3.0699 |
| | XGBoost | 0.2504 | 0.9999 | 8.1545 | 0.9011 | 2.717 | 3.01 |
| MSCFD | PLSR | 7.1537 | 0.9051 | 8.4846 | 0.8699 | 2.6113 | 2.8929 |
| | RF | 3.5235 | 0.9801 | 9.7311 | 0.8281 | 2.2768 | 2.5224 |
| | XGBoost | 0.016 | 0.9999 | 7.7869 | 0.8966 | 2.8452 | 3.1521 |
| LRFD | PLSR | 9.1212 | 0.8458 | 9.3074 | 0.8427 | 2.3804 | 2.6372 |
| | RF | 3.7104 | 0.9791 | 7.8726 | 0.8817 | 2.8143 | 3.1178 |
| | XGBoost | 0.5386 | 0.9995 | 8.0785 | 0.8976 | 2.7425 | 3.0383 |

Notes: R is the original spectrum merely using S-G smooth, LR is the logarithmic reciprocal, CR is the continuum removal, MSC is multivariate scattering correction, FD is the first-order differential, and CRFD, MSCFD and LRFD are the first-order differentials based on CR, MSC and LR, respectively. PLSR is partial least squares regression, RF is the random forest, and XGBoost is extreme gradient boosting.

The accuracy of the SOC content simulation in this study is similar to those of previous studies in the TRSR. Here, we report the $R^2$ ranging from 0.55 to 0.90 for verification sets in this study. Similarly, other studies report the r-value ranged from 0.06 to 0.99 [54], and $R^2$ ranged from 0.5 to 0.93 [55]. Additionally, the RMSE here is lower than the results of [54]. Therefore, the simulation results in this study are considered reliable.

### 3.4. Models Stacking Results and Feature Importance Analysis

In this study, stack models were designed to extract spectral features of SOC content by integrating different spectral transformation methods. After eight spectral transformation method data were learned by RF and XGBoost, these two machine learning models, the predicted results of 16 combinations were used to build the first three stack models (Stack1, Stack2, Stack3). According to Figure 5, in different stack models, the importance of various combinations is not exactly the same. However, FD-XG, CR-XG, and CRFD-XG are of high importance in all models, and FD-RF has a significant contribution in Stack3 models. Furthermore, the input variables of the models will be optimized based on the threshold of feature importance of the first-step three stack models (importance > 0.05), so as to establish the next step three stack models (Stack4, Stack5, Stack6).

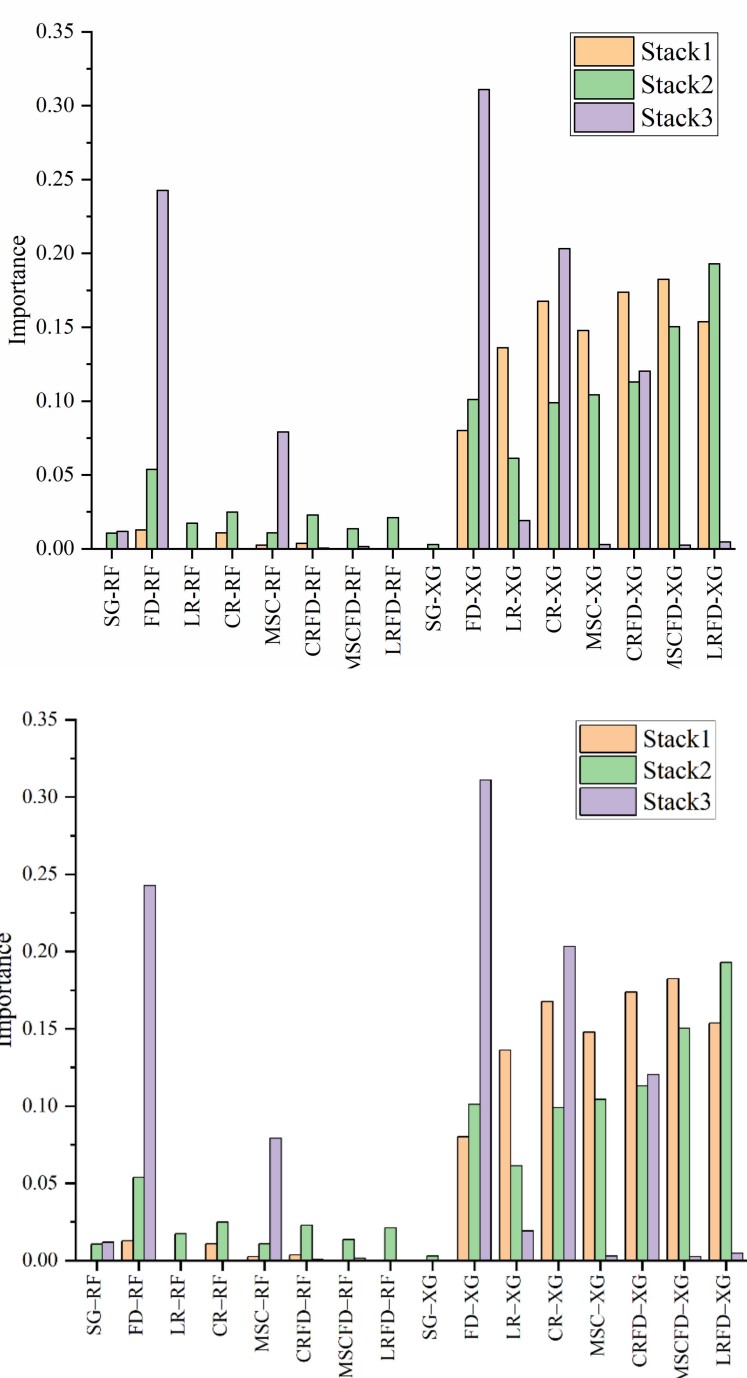

**Figure 5.** Feature importance of Stack1, Stack2, and Stack3 (Stacking models with SVR, RF, XG (i.e., XGBoost) and 16 pretreatment-models, respectively).

Among the six stack models, Stack 6 showed the best simulation performance (RMSE = 7.3511, $R^2$ = 0.8963, RPD = 3.0139, RPIQ = 3.339) (Table 5). Besides, compared to the results of Figure 6, the ensemble learning methods obtained higher accuracy than the single model and transformations. Especially after input variables were selected, the accuracy of models improved; for example, Stack 5 and Stack 6 both achieved a better modeling accuracy than that of Stack 2 and Stack 3. However, Stack 4 had lower accuracy than Stack 1.

**Table 5.** Accuracy comparison of six stack models.

| Models | Modeling Sets | | Verification Sets | | | |
|---|---|---|---|---|---|---|
| | RMSE | R² | RMSE | R² | RPD | RPIQ |
| Stack1 (16 combinations + SVR) | 0.3757 | 0.9998 | 7.4795 | 0.8968 | 2.9622 | 3.2817 |
| Stack2 (16 combinations + RF) | 0.5313 | 0.9996 | 7.5067 | 0.8939 | 2.9514 | 3.2698 |
| Stack3 (16 combinations + XGBoost) | 0.0743 | 1.0000 | 7.7530 | 0.8807 | 2.8577 | 3.1659 |
| Stack4 (7 selected combinations + SVR) | 1.7979 | 0.9940 | 7.9486 | 0.8893 | 2.7873 | 3.0880 |
| Stack5 (8 selected combinations + RF) | 0.3987 | 0.9997 | 7.4721 | 0.8953 | 2.9651 | 3.2849 |
| Stack6 (5 selected combinations + XGBoost) | 0.9194 | 0.9987 | 7.3511 | 0.8962 | 3.0139 | 3.3390 |

Notes: Stack4 represented FD-XGBoost, LR-XGBoost, CR-XGBoost, MSC-XGBoost, CRFD-XGBoost, MSCFD-XGBoost, LRFD-XGBoost combinations + SVR; Stack5 represented FD-XGBoost, FD-RF, LR-XGBoost, CR-XGBoost, MSC-XGBoost, CRFD-XGBoost, MSCFD-XGBoost, LRFD-XGBoost combinations + RF; Stack6 represented FD-RF, MSC-RF, FD-XGBoost, CR-XGBoost, CRFD-XGBoost combinations + XGBoost. These are among 16 combinations represented RF and XGBoost machine learning models, combined with eight spectral transformations (S-G, LR, CR, MSC, FD, LRFD, CRFD, and MSCFD).

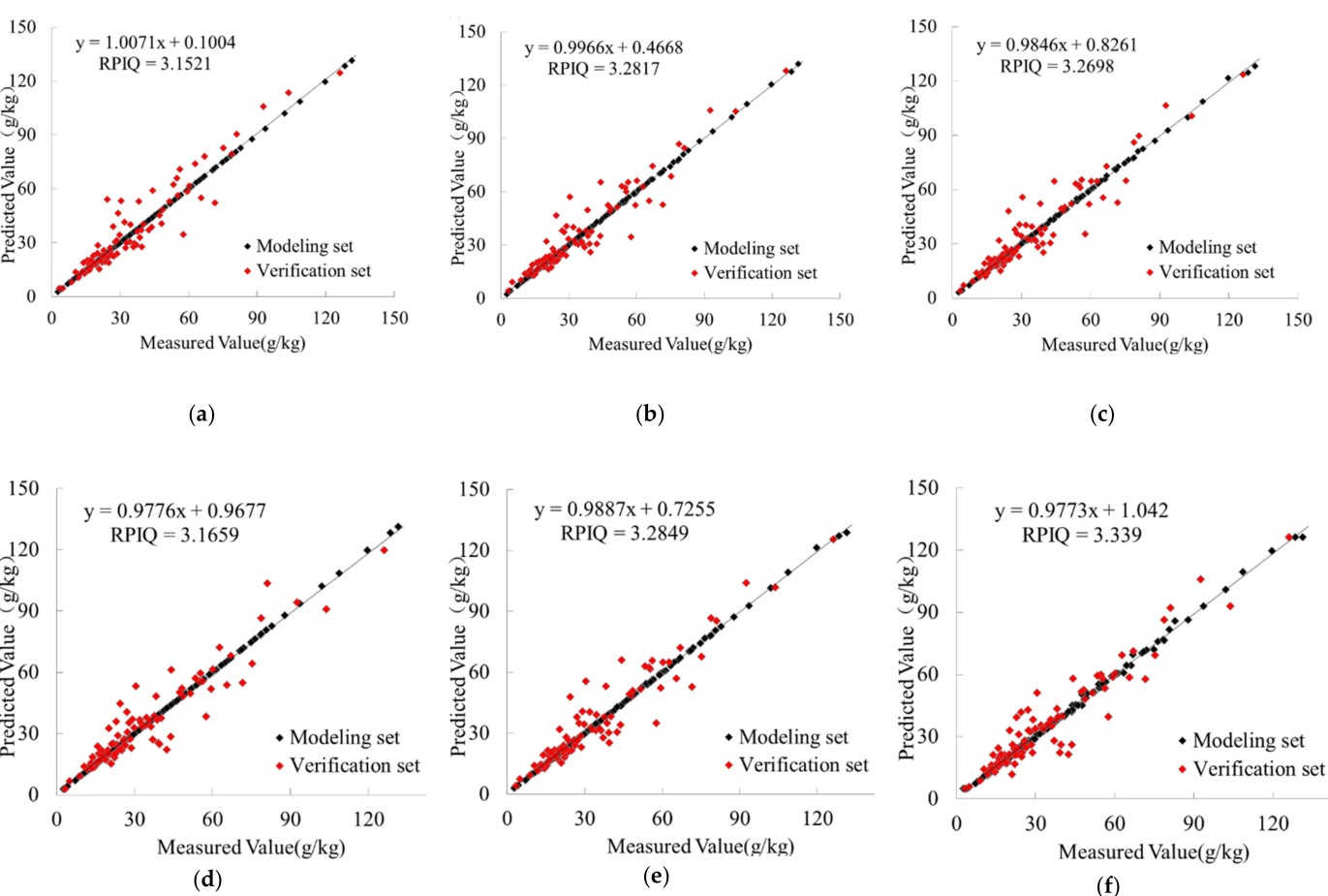

**Figure 6.** Scatter diagrams of six representative prediction models. (**a**) MSCFD–XG, (**b**) Stack 1, (**c**) Stack 2, (**d**) Stack 3, (**e**) Stack 5 and (**f**) Stack 6.

## 4. Discussion

### 4.1. Combine of Pre-Processing Transformations and Models in SOC Simulation

In some past SOC simulation studies in the alpine ecosystem area (Table 6), S-G, FD, and LR are the most commonly used pre-processing methods to improve the prediction accuracy of soil organic carbon, while RF, SVM, and PLSR were popular model algorithms.

**Table 6.** Comparison of SOC simulation accuracy in an alpine ecosystem.

| Reference | Depth | Pre-Processing Transformations | Models | $R^2$ | RPIQ | RPD |
|---|---|---|---|---|---|---|
| Yang et al., 2015 | 0–20 | FD, SD, LR, BD | PLS, BP neural network | 0.01~0.96 | | |
| Chen et al., 2019 | 0–20 | SG, log(1/R) | PLS, MLP | 0.80~0.92 | 2.3~3.68 | |
| Xiao et al., 2020 | 0–20 | SG, FD | PLSR | 0.62~0.79 | | 1.6~2.2 |
| Ogrič, M et al., 2019 | 0–20 | FD, SD | PLSR | 0.78~0.95 | 0.47~5.31 | 0.41~4.74 |
| Wei, et al., 2020 | 0–100 | MSC, MC, MA, SG, SG-FD, SG-SD, FD, SD, CR, LR | RR, KRR, BRR, AdaBoost | 0.58~0.91 | | |
| This study, 2022 | 0–20 | SG, MSC, LR, CR, FD, MSCFD, LRFD, CRFD | PLS, RF, SVR, XGBoost | 0.51–0.90 (single model); 0.88~0.90 (stack models) | 1.84~3.15; 3.17~3.34 | 1.37~2.85; 2.86~3.01 |

However, the difference in simulation accuracy between different spectral transformation methods and model algorithms is quite large [56]. For example, Chen et al. (2019) used MLP and PLS models combined with S-G smooth to predict SOC content based on 547 soil samples, and the $R^2$ is 0.92 and 0.8, respectively [57]. But in our study, the simulation accuracy of SOC content based on S-G smoothing is much lower than that of FD, LR, CR, MSCFD, LRFD, and CRFD spectral transformations (Table 4). Meanwhile, the optimal spectral pre-processing and simulation models are different for different soil types; for example, the study of Yang et al. (2015) [54] in Three-Rivers Source and Xiao et al. 2020 [55] in the northeast of Tibet. For different research regions, although these two regions both belong to the continental subarctic climate, the simulation accuracies are also different (Ogrič et al. 2019) [58], $R^2$ is 0.78 and 0.95, respectively.

In this paper, through the meta-learning of stacking method, six integrated models integrating various pretreatment methods and regression models were built to fully combine the advantages of various methods to mine the spectral features of data. However, due to the limited number of samples collected, the established models have some over-fitting phenomenon of training sets, and the RMSE error of validation sets is in the concentration of 7.5–8.0 g/kg in general. Therefore, in future studies, on the one hand, more sampling sites will be collected, especially in the western and northern regions of TRSR; then, different eco-environment regions will be divided for the whole TRSR.

On the other hand, different models will be built based on different eco-environment regions, soil types, vegetation types, or SOC content gradients. Last, in order to extract the characteristic information of different soil samples, it is worthwhile to integrate more pretreatment methods, such as orthogonal signal correction (OSC) [59].

### 4.2. Spatial Differentiation of SOC Content

The results of this study indicated that the spatial distribution of SOC content in the TRSR varied due to several factors. In the vertical direction, SOC content increased with the increase of altitude. This may be explained by the observation that areas of high altitude with low temperature are associated with reduced SOC decomposition. Additionally, periodic grazing may also be alleviated at higher altitudes, which may lead to an increase in SOC input. In addition, SOC content decreased with increasing depth. This decrease likely occurs because the humus in the surface soil is abundant, resulting in stronger biological activity and richer SOC content. With the deepening of the soil layer, the distribution of animal and plant material is reduced, and the SOC content is less. Further, in the deep soil, the increase of gravels is not conducive to the long-term storage of organic matter. Via these proposed mechanisms, climate, terrain conditions and human interference all affected vegetation growth and respiration, which further affect soil carbon input and SOC content [60,61]

SOC content of different vegetation types varied greatly (Table 7). The content of SOC in swamp meadow was the highest (98.91 g/kg), while SOC content in desert grassland was

the lowest (5.07 g/kg). Correspondingly, the above-ground biomass of swamp meadow was also significantly higher than that of desert grassland in this study (390.93 g/m$^2$ vs. 56.42 g/m$^2$). Because of this, the above-ground biomass is one of the main reasons affected SOC content [62]. Except for swamp meadow, woodland and farmland, the variation coefficient of SOC content displayed moderate variation, indicating that the SOC content spatial distribution exhibits obvious spatial heterogeneity.

**Table 7.** SOC content statistics of each vegetation type.

| Sample Type | Sample Size | Average (g/kg) | Standard Deviation (g/kg) | Kurtosis | Skewness | Coefficient of Variation (%) |
|---|---|---|---|---|---|---|
| Overall | 272 | 34.6800 | 22.9200 | 3.2000 | 1.6400 | 66.1000 |
| Alpine meadow | 101 | 43.8963 | 20.6124 | 0.3268 | 0.6335 | 46.9570 |
| Alpine grassland | 36 | 21.9632 | 9.6188 | 1.1223 | 1.1721 | 43.7952 |
| Alpine meadow grassland | 46 | 31.8256 | 15.3990 | 1.5478 | 1.2023 | 48.3856 |
| Swamp meadow | 8 | 98.9051 | 28.7973 | −1.8473 | −0.1171 | 29.1161 |
| Temperate grassland | 59 | 23.1995 | 11.7917 | 31.8100 | 5.0835 | 50.8274 |
| Desert grassland | 6 | 5.0734 | 2.6793 | −1.6993 | 0.4736 | 52.8112 |
| Farmland | 13 | 18.1501 | 4.0150 | −0.9805 | −0.6302 | 22.1209 |
| Woodland | 3 | 45.5200 | 11.6000 | - | 1.7200 | 25.4800 |

### 4.3. Soil Carbon Change Analysis across Tibetan Plateau

Previous studies have indicated that the mean topsoil SOC turnover duration is 338 years across Tibetan Plateau (TP) grassland sites [63], and have also highlighted the fact that there is a significant spatial correlation between the turnover time and MAP (mean annual precipitation). Specifically, increased precipitation is strongly associated with a shorter SOC turnover time in TP grassland sites. Negative correlations between SOC turnover time and temperature in the TP are linked to an increase in microbial decomposition with temperature [64]. Moreover, regional-scale studies observed that 57.4% of spatial variation of the SOC turnover time was explained by MAP and altitude in TP grassland.

As a key factor driving ecosystem dynamics, temperatures over the past decades have been increasing at a faster rate in the TP than in the Northern hemisphere on average. Zhai et al. (2005) found that precipitation increased significantly in the southeastern TP from 1950 to 2008 [65]. At the same time, the social economy of the TP has developed continuously since the 1950s [66]. For example, since 2001, the total fence-closed area and forbidden grazing area both increased drastically [67], while at the same time, the livestock number increased from 5,700,000 to 6,670,000 based on the Qinghai province statistical yearbook (http://tjj.qinghai.gov.cn/2021.12.10, accessed on 12 October 2021).

These trends are predicted to lead to more serious overgrazing in the TP [68], and overgrazing leads to the decrease of grass biomass, as found in our previous studies in Xinghai County [14]. Xinghai County is also the sampling point distribution area in this study, where the warm-wet climate conditions are conducive to improving plant growth, but the increase of population and livestock leads to a decrease in grass biomass. On the other hand, overgrazing may therefore accelerate carbon decomposition rates and shorten the turnover time of SOC by shifting soil microbial composition from fungi-dominated to bacterial-dominated communities [28]. In this way, the biomass C storage and soil C stock decrease with increasing grazing intensity [29].

Above all, a warming climate can accelerate the microbial breakdown of organic carbon and lead to the increased release of carbon dioxide, thereby causing a feedback cycle that can accelerate climate warming [26]. On the other hand, a warming climate may improve the net primary productivity and increase soil carbon sequestration in some

areas [27]. The balance of these fluxes determines the nature of the Alpine ecosystem feedback to warming. Therefore, SOC content monitoring is vital for the estimation of the soil carbon sink along with its feedback on climate change.

## 5. Conclusions

Due to climate warming and intensified human intervention, SOC decomposition accelerated in the alpine ecosystem. Therefore, the high-accuracy simulation of SOC content is vital for the carbon sink estimation and its response to climate change. In this study, a total of 272 samples were collected in the Three-Rivers Source Regions. The ensemble learning method by the stack models was conducted to improve the estimation accuracy of SOC content. Stack models were designed to extract spectral features of SOC by ensemble different spectral transformation methods. After data from eight spectral transformation methods were learned by RF and XGBoost, and the predicted results of 16 combinations were used to build the first-step stack models (Stack1, Stack2, Stack3). Then, the input variables of the model will be optimized based on the threshold of feature importance of the first-step three stack models (importance > 0.05)so as to establish the next step three stack models (Stack4, Stack5, Stack6). The SOC content estimation was also conducted using the single model and eight spectral transformation data. The results in this study showed that the ensemble learning methods obtained higher accuracy than the single model and transformations. Especially after selecting the input variables, the accuracy of the stack models significantly improved compared to the 16 combination variables. Among the six stack models, Stack 6 (FD-RF, MSC-RF, FD-XG, CR-XG, CRFD-XG 5 combinations + XGBoost) showed the best simulation performance (RMSE = 7.3511, $R^2$ = 0.8963, RPD = 3.0139, RPIQ = 3.339), and obtained higher accuracy than Stack3(16 combinations + XGBoost) performance (RMSE = 7.7530, $R^2$ = 0.8807, RPD = 2.8577, RPIQ = 3.1659).Overall, our results suggest the ensemble learning method of spectral transformations and simulation models could be used to estimate the SOC content. This study can provide a useful reference for high-precision estimation and monitoring of SOC content in alpine ecosystems in the future.

**Author Contributions:** Conceptualization, W.Z.; methodology, W.Z. and L.X.; Software, H.L.; validation, H.L. and S.W.; investigation, W.Z., H.L., L.X., Y.T., T.W. and W.Y.; writing, W.Z. and H.L.; funding acquisition, W.Z. All authors have read and agreed to the published version of the manuscript.

**Funding:** This work and article processing charge were funded by the Project of Chongqing Science and Technology Bureau (cstc2021jcyj-msxmX0384, cstc2019jscx-fxydX0036), the Postdoctoral Start-Up Project of Southwest University (SWU020015) the National Natural Science Foundation of China. (41930647, 41501575, 41977337), the Innovation Project of LREIS (O88RA600YA). We thank AJE for their linguistic assistance during the preparation of this manuscript.

**Institutional Review Board Statement:** Not Applicable.

**Informed Consent Statement:** Not Applicable.

**Data Availability Statement:** The data used to support the findings of this study are available from the corresponding author upon request.

**Acknowledgments:** The authors are thankful for the support of the staff members of the Qinghai Ecological Environment Monitoring Center and Qinghai University during the field and lab work in the TRSR. We also thank Shannon Elliot at Michigan State University for his assistance with the English language and grammatical editing.

**Conflicts of Interest:** There is no conflict of interest between co-authors in this article.

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
