# Peer review of "Simulation of Soil Organic Carbon Content Based on Laboratory Spectrum in the Three-Rivers Source Region of China"

_remotesensing, doi:10.3390/rs14061521_

Round 1

Reviewer 1 Report

I'm reviewing "Simulation of soil organic carbon content based on laboratory spectrum in the Three-Rivers Source Region of China".
The proposed manuscript is well written and clearly exposed.
I agree with the authors on the convenience of a further spatial amplitude in soil sampling. The processing flows comparison is neverless sound.

 the following minor notices.

I report a typo at line [15-16]:
"of spectral transformations to 15 accurately to simulate SOC"

Part of the manuscript (e.g. table2, lines 368-370) suffer of some kind of layout issues.

Reviewer 2 Report

The authors have presented the work titled "Simulation of soil organic carbon content based on laboratory spectrum in the Three-Rivers Source Region of China"

I really commend the effort of the authors for this good work.

Few comments

Why was the predicted maps of the different ensemble models not presented? This is because it will be nice to see the visual representation of the different models.

Also, I recommend that the following studies be included in the literature

  1. Biney, J.K.M.; Saberioon, M.; Borůvka, L.; Houška, J.; Vašát, R.; Chapman Agyeman, P.; Coblinski, J.A.; Klement, A. Exploring the Suitability of UAS-Based Multispectral Images for Estimating Soil Organic Carbon: Comparison with Proximal Soil Sensing and Spaceborne Imagery. Remote Sens. 202113, 308. https://doi.org/10.3390/rs13020308
  2. Biney, J. K. M., Blöcher, J. R., Borůvka, L., & Vašát, R. (2021). Does the limited use of orthogonal signal correction pre-treatment approach to improve the prediction accuracy of soil organic carbon need attention?. Geoderma388, 114945.
  3. JOHN, K.; Abraham Isong, I.; Michael Kebonye, N.; Okon Ayito, E.; Chapman Agyeman, P.; Marcus Afu, S. Using Machine Learning Algorithms to Estimate Soil Organic Carbon Variability with Environmental Variables and Soil Nutrient Indicators in an Alluvial Soil. Land 20209, 487. https://doi.org/10.3390/land9120487

Good luck!

Reviewer 3 Report

It is an interesting paper because of the area in which it takes place, the permafrost, rare and well-known. It has a good statistical development

The language should be checked and the wording taken care of, so in Table 6 it says simulaiton?

Line 131: Indicate the type of climate in the Köppen Geiger classification.

The bibliography is excessive, some repetitive and very biased in terms of its production.

References should be removed. 

Round 2

Reviewer 3 Report

The paper has improved with corrections. It's OK now

Author Response

Dear professor,

Thank you for your valuable comments on our manuscript and your recognition of our research work.

with regards

yours Wei Zhou